

# Fermionic Bogomol'nyi-Prasad-Sommerfield Wilson loops in four-dimensional $\mathcal{N} = 2$ superconformal gauge theories

Hao Ouyang[1*] and Jun-Bao Wu[2,3†]

**1** Center for Theoretical Physics and College of Physics,
Jilin University , Changchun 130012, P. R. China
**2** Center for Joint Quantum Studies and Department of Physics, School of Science ,
Tianjin University, 135 Yaguan Road, Tianjin 300350, P. R. China
**3** Peng Huanwu Center for Fundamental Theory, Hefei, Anhui 230026, P. R. China

* haoouyang@jlu.edu.cn ,   corresponding author:   † junbao.wu@tju.edu.cn

## Abstract

We construct for the first time Drukker-Trancanelli (DT) type fermionic Bogomol'nyi-Prasad-Sommerfield (BPS) Wilson loops in four-dimensional $\mathcal{N} = 2$ superconformal $SU(N) \times SU(N)$ quiver theory and $\mathcal{N} = 4$ super Yang-Mills theory. The connections of these fermionic BPS Wilson loops have a supermatrix structure. We construct timelike BPS Wilson lines in Minkowski spacetime and circular BPS Wilson loops in Euclidean space. These Wilson loops involve dimensionful parameters. For generic values of parameters, they preserve one real (complex) supercharge in Lorentzian (Euclidean) signature. Supersymmetry enhancement for Wilson loops happens when the parameters satisfy certain constraints. The nature of such loops is quite different from the Wilson loop operators involving fermions constructed previously in the literature on four-dimensional gauge theories. We hope that further investigations of such new Wilson loops will explore deep structures in both the gauge theories and gauge/gravity dualities.

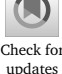

# 1 Introduction

Line operators are very important in the study of gauge theories. The vacuum expectation values (vevs) of Wilson-'t Hooft line operators can be used to distinguish different (infrared) phases of gauge theories [1,2]. The precise description of the gauge theory should take into account the choice of the set of Wilson-'t Hooft line operators included in the theory [3]. Also notice that Wilson lines and 't Hooft lines can carry charges of 1-form global symmetries [4].

In supersymmetric gauge theories, line operators preserving part of the supersymmetries constantly attract much attention [5,6]. In $\mathcal{N} = 4$ super Yang-Mills theory (SYM), for a Wilson line along a timelike straight line or a Wilson loop along a circular loop in Euclidean theories to be supersymmetric, the line/loop should also couple to the scalar fields in the theory [7–9]. The former can be understood as the dimensional reduction of a lightlike Wilson line in ten-dimensional $\mathcal{N} = 1$ SYM.

Such Bogomol̀nyi-Prasad-Sommerfield (BPS) Wilson loops (WLs) also play an important role [7,8] in AdS/CFT correspondence since the earlier days of this holographic duality [10–12]. The vev of a circular half-BPS Wilson loop depends on the SYM coupling constant nontrivially. It was conjectured that this vev can be computed by using a Gaussian matrix model [13]. The result in the large $N$ and large 't Hooft coupling limit is consistent with the prediction from the IIB superstring theory on $AdS_5 \times S^5$ background [9,14]. This is one of the first precise checks of the AdS/CFT correspondence beyond checks related to various non-renormalization theorems [15–17]. Later, the conjecture about the reduction to the Gaussian matrix model was proved using supersymmetric localization [18]. People have found a large amount of BPS Wilson loops with fewer supersymmetries in $\mathcal{N} = 4$ SYM. Among them, there are Zarembo loops [19] and Drukker-Giombi-Ricci-Trancanelli (DGRT) loops [20,21]. The special property of Zarembo loops is that their vev are constants protected by supersymmetries. As for DGRT loops, though they are also BPS, their vevs depend on the SYM coupling constant. Further classification was discussed in [22].

The situation for BPS Wilson loops in three-dimensional super-Chern-Simons theories is more complicated and interesting. The construction of bosonic BPS WLs [23–26] here is quite similar to the one in the four-dimensional SYM when the auxiliary fields in the vector multiplets are used. Taking into account the equations of motion, these auxiliary fields will be replaced by scalar bilinears. Since the scalars in three-dimensional spacetime have dimension one-half. The coefficients of these bilinears are dimensionless and the BPS condition imposes constraints on these coefficients. More or less surprisingly, this Gaiotto-Yin type BPS WL along a straight line or a circular loop in ABJM theory can be at most 1/6-BPS. However, the study of probe F-strings/M2-branes in the string/M-theory side predicts that there should exist half-BPS Wilson loops [24,26]. This puzzle was resolved by Drukker and Trancanelli through the

construction of half-BPS fermionic Wilson loops [27]. A key point here is to include fermions in the construction of the super-connection. Since the fermions in three-dimensional space-time have dimension one, we still do not need to introduce dimensionful parameters in the construction. Later, fermionic 1/6-BPS Wilson loops were found [28, 29], and these loops are interpolation between bosonic 1/6-BPS Wilson loops and fermionic half-BPS Wilson loops. These 1/6-BPS Wilson loops are not locally $SU(3)$ invariant, and this indicates that they are not dual to F-strings with Dirichlet boundary conditions in $\mathbf{CP}^3$. In fact, they are dual to F-strings with complicated mixed boundary conditions [30]. These general 1/6-BPS Wilson loops can be thought of as marginal deformations of half-BPS Wilson loops from the defect conformal field theory (dCFT) point of view. Although the marginality of the deformations is yet to be proved at quantum level on the field theory side, it is supported by the general classification of superconformal line defects and the studies of their deformations [31] and the fact that there are massless fermions on the worldsheet of F-string dual to the half-BPS Wilson loop [30, 32, 33]. General BPS Wilson loops in $\mathcal{N} \geq 2$ super-Chern-Simons theories were constructed in [34] based on [28, 29]. The moduli spaces of such loops were shown to be quiver varieties [35]. For many important aspects of Wilson loops in three-dimensional super-Chern-Simons theories, we would like to recommend the wonderful review [36].

We now review some features of the fermionic BPS WLs in three-dimensional theories. Two features will be compared with four-dimensional counterparts, and third one is about the pattern in the construction which also appears in the four-dimensional case. The first feature is above the supersymmetry enhancement. Using ABJM theory as an example, the fermionic BPS WLs in [28, 29] preserve at least the same supercharges as the bosonic BPS WLs when they are along the same timelike straight lines or circular loops, and the fermionic WLs can have enhancement supersymmetries when the parameters in these WLs take certain special values. The second is about the choice between the trace and the supertrace. Although a super-connection was used, it was found that one should use a trace instead of a supertrace in the construction of fermionic BPS Wilson loops [27]. Later such WLs were rewritten using a supertrace accompanied by adding certain shifts [35–37]. Also notice that the earlier construction of fermionic WLs with fewer supersymmetries in [38] used a supertrace accompanied by multiplying a constant twist matrix. It is interesting to study switching between the above two new approaches. The third feature is about a pattern in the construction of fermionic super-connections. The construction of fermionic BPS WLs is more complicated than the bosonic ones. The superconnection $L_f$ of a fermionic BPS WL can be constructed by suitable deformation of a bosonic BPS WL with connection $L_B$ [28, 29, 34]. In [35–37], it was noticed the following pattern in such deformation, $L_f = L_B + (\cdots)QG + (\cdots)G^2$ with $Q$ one of the supercharges preserved by $L_B$ and $G$ certain linear combination of scalar fields. $(\cdots)$'s are coefficients to be determined.

It is natural to explore whether one can construct BPS fermionic WLs in four-dimensional superconformal gauge theories. In this paper, we successfully construct WLs in a simple four-dimensional $\mathcal{N} = 2$ quiver superconformal theory and $\mathcal{N} = 4$ SYM. Our construction can be easily generalized to general quiver superconformal theories. For the fermionic WLs to be BPS, we should also introduce scalar bilinears besides the scalar terms already in the bosonic WLs. Simple dimensional analysis shows that we need to introduce dimensionful parameters in the construction. So the BPS Wilson lines along a timelike straight line are not scale invariant. They only preserve Poincare supercharges.[1] As for Wilson loops along circular loops, they preserve some linear combinations of Poincare supercharges and superconformal charges. The BPS fermionic WLs along timelike straight lines or circular loops preserve only a small part of the supercharges preserved by the BPS bosonic WLs along the same lines or loops. This is

---

[1]WLs only preserving Poincare supercharges were already appearing in three-dimensional $\mathcal{N} = 2$ super-Chern-Simons theories when the matter fields have non-canonical dimensions [34].

quite different from the three-dimensional case as reviewed above. About the choice between the trace and the supertrace, we find that in the four-dimensional case, we should employ a supertrace in the approach used in [27]. For $\mathcal{N} = 4$ SYM, since there is only one node in its $\mathcal{N} = 2$ quiver diagram, we should employ more than one copy of the connection in the bosonic BPS WL. Similar construction involving multiple copies has been employed in [34]. As in many three-dimensional cases, The construction in the current paper is based on the pattern $L_f = L_B + (\cdots)QG + (\cdots)G^2$ mentioned above.

This paper provides the first construction of Drukker-Trancanelli type fermionic BPS Wilson loops. This construction is quite different from the Wilson loops involving fermions in the four-dimensional gauge theory literature [9, 39–41]. There are many aspects of these novel fermionic loops to be explored. Such investigations should be important to further studies of both gauge theories and gauge/gravity dualities. We leave these investigations to further work and list some possible future directions in the discussion section.

The paper is organized as follows. In section 2, we construct fermionic BPS Wilson loops in $\mathcal{N} = 2$ superconformal $SU(N) \times SU(N)$ quiver theory. In section 3, we construct fermionic BPS Wilson loops in $\mathcal{N} = 4$ SYM. The last section is dedicated to conclusions and discussion. We summarize our conventions in appendix A. Appendix B contains some technical details.

## 2 Fermionic BPS Wilson loops in $\mathcal{N} = 2$ superconformal $SU(N) \times SU(N)$ quiver theory

In this section, we introduce the $\mathcal{N} = 2$ superconformal $SU(N) \times SU(N)$ quiver theory which is a marginal deformation of the $\mathbb{Z}_2$ orbifold of $\mathcal{N} = 4$ SYM. A detailed discussion of the orbifold procedure can be found in [42]. Then we construct fermionic BPS Wilson loops along an infinite timelike straight line and a circle in Lorentzian and Euclidean signatures, respectively.

### 2.1 $\mathcal{N} = 2$ superconformal $SU(N) \times SU(N)$ quiver theory

We begin by introducing the $\mathcal{N} = 2$ superconformal $SU(N) \times SU(N)$ quiver theory which, as we have just mentioned, is a marginal deformation of the $\mathbb{Z}_2$ orbifold of the $\mathcal{N} = 4$ SYM. We use six-dimensional (6d) spinorial notations for the spinors. The fields in the two $\mathcal{N} = 2$ vector multiplets corresponding to two gauge group factors can be arranged into $2 \times 2$ block matrices:

$$A_\mu = \begin{pmatrix} A_\mu^{(1)} & 0 \\ 0 & A_\mu^{(2)} \end{pmatrix}, \quad \mu = 0, \ldots, 5,$$

$$\lambda_\alpha = \begin{pmatrix} \lambda_\alpha^{(1)} & 0 \\ 0 & \lambda_\alpha^{(2)} \end{pmatrix}, \quad \alpha = 1, 2, \tag{1}$$

where $A_m$ with $m = 0, \ldots, 3$ is the gauge field and $A_{4,5}$ are two real scalars. The $SO(1, 5)$ Weyl spinors $\lambda_1$ and $\lambda_2$ have chirality $-1$ for $\Gamma^{012345}$ and satisfy the reality condition $\bar{\lambda}^\alpha = -\epsilon^{\alpha\beta} \lambda_\beta^c$ where $\epsilon^{\alpha\beta}$ is the antisymmetric symbol with $\epsilon^{12} = 1$.[2] (See appendix A for our conventions on the spinors and gamma matrices). The matter content consists of two bifundamental hypermultiplets with component fields:

$$q^\alpha = \begin{pmatrix} 0 & q^{(1)\alpha} \\ q^{(2)\alpha} & 0 \end{pmatrix}, \qquad \psi = \begin{pmatrix} 0 & \psi^{(1)} \\ \psi^{(2)} & 0 \end{pmatrix}, \tag{2}$$

---

[2]Later, we will use $\epsilon_{\alpha\beta}$ which is defined to be the inverse of $\epsilon^{\alpha\beta}$, $\epsilon_{\alpha\beta} \epsilon^{\beta\gamma} = \delta_\beta^\alpha$.

where $q^{1,2}$ are complex scalars and $\psi$ is an $SO(1,5)$ Weyl spinor of chirality $+1$ for $\Gamma^{012345}$. We denote by $q_\alpha$ the complex conjugate of $q^\alpha$. The action of the $\mathcal{N}=2$ gauge theory is

$$
\begin{aligned}
S_{\mathcal{N}=2} \;=\; & \int d^4x \left( -\frac{1}{4}\mathrm{Tr}(F_{\mu\nu}F^{\mu\nu}) - \frac{i}{2}\mathrm{Tr}(\bar\lambda^\alpha \Gamma^\mu D_\mu \lambda_\alpha) - D_\mu q_\alpha D^\mu q^\alpha - i\bar\psi \Gamma^\mu D_\mu \psi \right. \\
& + \sqrt{2}g\,\bar\lambda^{\alpha A} q_\alpha T_A \psi - \sqrt{2}g\,\bar\psi T_A q^\alpha \lambda_\alpha^A - g^2(q_\alpha T^A q^\beta)(q_\beta T_A q^\alpha) \\
& \left. + \frac{1}{2}g^2(q_\alpha T_A q^\alpha)(q_\beta T^A q^\beta) \right),
\end{aligned}
\tag{3}
$$

where $T^A$ are the generators of the gauge group. The coupling constants for the two gauge group factors can be independently varied while preserving $\mathcal{N}=2$ superconformal symmetry. We assemble them into a matrix:

$$
g = \begin{pmatrix} g^{(1)} I_N & 0 \\ 0 & g^{(2)} I_N \end{pmatrix},
\tag{4}
$$

where we denote by $I_N$ the $N \times N$ identity matrix. At the orbifold point where $g^{(1)}=g^{(2)}$, the theory can be obtained via orbifolding $\mathcal{N}=4$ SYM by $\mathbb{Z}_2$. The definitions of the covariant derivatives are

$$
\begin{aligned}
D_\mu \lambda &= \partial_\mu \lambda - ig[A_\mu, \lambda], \tag{5} \\
D_\mu q^\alpha &= \partial_\mu q^\alpha - ig A_\mu q^\alpha, \tag{6} \\
D_\mu q_\alpha &= \partial_\mu q_\alpha + ig q_\alpha A_\mu, \tag{7} \\
D_\mu \Psi &= \partial_\mu \Psi - ig A_\mu \Psi, \tag{8}
\end{aligned}
$$

with $A_\mu = A_\mu^A T_A$. The definition of the field strength is

$$
F_{\mu\nu} = \partial_\mu A_\nu - \partial_\nu A_\mu - ig[A_\mu, A_\nu].
\tag{9}
$$

One can show that the action is invariant under the $\mathcal{N}=2$ superconformal transformations:[3]

$$
\begin{aligned}
\delta A_\mu &= -i\bar\xi^\alpha \Gamma_\mu \lambda_\alpha = i\bar\lambda^\alpha \Gamma_\mu \xi_\alpha, \\
\delta q^\alpha &= -i\sqrt{2}\bar\xi^\alpha \psi, \\
\delta q_\alpha &= -i\sqrt{2}\bar\psi \xi_\alpha, \\
\delta \lambda_\alpha^A &= \frac{1}{2}F_{\mu\nu}^A \Gamma^{\mu\nu}\xi_\alpha + 2ig q_\alpha T^A q^\beta \xi_\beta - ig q_\beta T^A q^\beta \xi_\alpha - 2A_a^A \Gamma^a \vartheta_\alpha, \\
\delta \bar\lambda^{\alpha A} &= -\frac{1}{2}\bar\xi^\alpha F_{\mu\nu}^A \Gamma^{\mu\nu} - 2ig q_\beta T^A q^\alpha \bar\xi^\beta + ig q_\beta T^A q^\beta \bar\xi^\alpha + 2\bar\vartheta^\alpha A_a^A \Gamma^a, \\
\delta \psi &= -\sqrt{2}D_\mu q^\alpha \Gamma^\mu \xi_\alpha - 2\sqrt{2}q^\alpha \vartheta_\alpha, \\
\delta \bar\psi &= \sqrt{2}\bar\xi^\alpha \Gamma^\mu D_\mu q_\alpha - 2\sqrt{2}\bar\vartheta^\alpha q_\alpha,
\end{aligned}
\tag{10}
$$

where $\xi_\alpha = \theta_\alpha + x^m \Gamma_m \vartheta_\alpha$ and the index $a = 4,5$. The constant spinors $\theta_\alpha$ and $\vartheta_\alpha$ generate Poincaré supersymmetry transformations and conformal supersymmetry transformations, respectively.

## 2.2 BPS Wilson lines in Minkowski spacetime

Let us start with reviewing the construction of bosonic BPS WLs. In Minkowski spacetime, one can define a 1/2 BPS Wilson line along the timelike infinite straight line $x^m = \delta_0^m \tau$ as

$$
W_{\mathrm{bos}} = \mathcal{P}e^{i\int d\tau L_{1/2}(\tau)}, \quad L_{1/2} = gA_0 - gA_5.
\tag{11}
$$

---

[3]A typo in [42] has been corrected.

The persevered supersymmetries can be parameterized by $\xi_\alpha$ satisfying

$$\Gamma_5\Gamma_0\xi_\alpha = \xi_\alpha. \tag{12}$$

In three-dimensional $\mathcal{N} = 2$ superconformal Chern-Simons theories fermionic BPS Wilson lines can be constructed as deformations of the bosonic BPS ones [36]. We follow a similar procedure here. Let us consider the Wilson line operator

$$W_{\text{fer}} = \mathcal{P}e^{i\int d\tau L}, \tag{13}$$

where the connection $L$ is a supermatrix analogous to the ones constructed in [27]:

$$L = L_{1/2} + B + F. \tag{14}$$

The matrices $B$ and $F$ are defined as

$$B = \begin{pmatrix} B^{(1)} & 0 \\ 0 & B^{(2)} \end{pmatrix}, \tag{15}$$

$$F = \zeta^c\psi + \bar{\psi}\eta, \tag{16}$$

$$\zeta^c = \begin{pmatrix} \zeta^{(1)c}I_N & 0 \\ 0 & \zeta^{(2)c}I_N \end{pmatrix}, \tag{17}$$

$$\eta = \begin{pmatrix} \eta^{(2)}I_N & 0 \\ 0 & \eta^{(1)}I_N \end{pmatrix}, \tag{18}$$

where $B^{(1)}$ and $B^{(2)}$ are products of scalar fields and $\zeta^c$ and $\eta$ are bosonic spinors. Different from the three-dimensional case, the parameters $\zeta^c$ and $\eta$ have the dimension of inverse square root of mass. We would like to construct a Wilson line invariant under a given supersymmetry transformation $\delta_\xi$ parameterized by $\xi_\alpha = \theta s_\alpha$ where $\theta$ is a real Grassmann variable and $s_\alpha$ are bosonic spinors. At this moment we restrict our attention to Poincaré supersymmetries and thus $s_\alpha$ are constant spinors along the line. It is convenient to define the preserved supercharge $Q_s$ as $\delta_\xi = \sqrt{2}\theta Q_s$. We need the transformation

$$\begin{aligned}
Q_s q^\alpha &= -i\bar{s}^\alpha\psi, \\
Q_s q_\alpha &= i\bar{\psi}s_\alpha, \\
Q_s\psi &= -D_\mu q^\alpha\Gamma^\mu s_\alpha, \\
Q_s\bar{\psi} &= \bar{s}^\alpha\Gamma^\mu D_\mu q_\alpha.
\end{aligned} \tag{19}$$

To make the Wilson line preserve supercharge $Q_s$ with a fixed $s_\alpha$ satisfying $s_\alpha = \Gamma_5\Gamma_0 s_\alpha$, we require $L$ to transform as [27,43]

$$Q_s L = \mathcal{D}_0 G_s \equiv \partial_0 G_s - i[L_{1/2} + B, G_s] + i\{F, G_s\}, \tag{20}$$

for some bosonic matrix $G_s$. Splitting this constraint into a fermonic and bosonic part, we find

$$Q_s B = i\{F, G_s\}, \tag{21}$$

$$Q_s F = \partial_0 G_s - i[L_{1/2} + B, G_s]. \tag{22}$$

Acting $Q_s$ on $F$, we get

$$Q_s F = -\zeta(D_\mu q^\beta\Gamma^\mu s_\beta) + (\bar{s}^\beta\Gamma^\mu D_\mu q_\beta)\eta. \tag{23}$$

In order that $Q_s F$ takes the form of $\partial_0 G_s + \dots$, we require

$$\Gamma_5\Gamma_0\eta = \eta, \; \zeta^c\Gamma_5\Gamma_0 = -\zeta^c, \tag{24}$$

and thus

$$Q_s F = \partial_0 G_s - i[L_{1/2}, G_s],\tag{25}$$

where

$$G_s = \zeta^c \Gamma_0 s_\alpha q^\alpha - q_\alpha \bar{s}^\alpha \Gamma_0 \eta.\tag{26}$$

Comparing equation (22) with (25), we need $[B, G_s] = 0$. Since $Q_s B = i\{F, G_s\}$, $B$ should be a sum of scalar bilinears. From $[B, G_s] = 0$, one can show that $B = -kG_s^2$ where $k$ is a complex number. Acting $Q_s$ on $G_s$, we find

$$Q_s G_s = -i\zeta^c \Gamma_0 s_\alpha \bar{s}^\alpha \psi - i\bar{\psi} s_\alpha \bar{s}^\alpha \Gamma_0 \eta = -\frac{i}{2}\bar{s}^\alpha \Gamma_0 s_\alpha \zeta^c \psi - \frac{i}{2}\bar{\psi}\eta \bar{s}^\alpha \Gamma_0 s_\alpha = -\frac{i}{2}\bar{s}^\alpha \Gamma_0 s_\alpha F.\tag{27}$$

Therefore $k = 2/(\bar{s}^\alpha \Gamma_0 s_\alpha)$ and the super-connection can be written as[4]

$$L = L_{1/2} + \frac{2i}{(\bar{s}^\alpha \Gamma_0 s_\alpha)} Q_s G_s - \frac{2}{(\bar{s}^\alpha \Gamma_0 s_\alpha)} G_s^2.\tag{28}$$

As we can see, $F$ is proportional to $Q_s G_s$ and $B$ is proportional to $G_s^2$. This pattern was first noticed in [36] for general fermionic BPS WLs in ABJM theory and later utilized in in [35,37]. A similar pattern will also appear in the construction of the circular WLs in this $\mathcal{N} = 2$ theory and the WLs in $\mathcal{N} = 4$ SYM later in this paper.

The next task is to know whether the Wilson loop can preserve other supercharges. We need to solve all the $u_\alpha$ satisfying $u_\alpha = \Gamma_5 \Gamma_0 u_\alpha$ and $Q_u L = \mathcal{D}_0 G_u$ where $G_u = \zeta^c \Gamma_0 u_\alpha q^\alpha - q_\alpha \bar{u}^\alpha \Gamma_0 \eta$. We need

$$[G_s^2, G_u] = 0,\tag{29}$$

$$\{Q_u G_s, G_s\} = \{Q_s G_s, G_u\}.\tag{30}$$

It follows from (29) that

$$G_u = K G_s = \begin{pmatrix} k^{(1)} I_N & 0 \\ 0 & k^{(2)} I_N \end{pmatrix} G_s,\tag{31}$$

or equivalently

$$\zeta^c \Gamma_0 u_\alpha q^\alpha - q_\alpha \bar{u}^\alpha \Gamma_0 \eta = K(\zeta^c \Gamma_0 s_\alpha q^\alpha - q_\alpha \bar{s}^\alpha \Gamma_0 \eta).\tag{32}$$

Then (30) leads to

$$\begin{aligned}
&\{Q_u G_s, G_s\} = \{Q_s G_s, G_u\} \\
\Leftrightarrow\ & Q_u G_s G_s = Q_s G_s G_u \quad \text{and} \quad G_s Q_u G_s = G_u Q_s G_s \\
\Leftrightarrow\ & Q_u G_u G_s = K Q_s G_s K G_s \quad \text{and} \quad G_s Q_u G_u = K G_s K Q_s G_s \\
\Leftrightarrow\ & (\bar{u}^\alpha \Gamma_0 u_\alpha) Q_s G_s = (\bar{s}^\beta \Gamma_0 s_\alpha) k^{(1)} k^{(2)} Q_s G_s \\
\Leftrightarrow\ & k^{(1)} k^{(2)} = \frac{\bar{u}^\alpha \Gamma_0 u_\alpha}{\bar{s}^\alpha \Gamma_0 s_\alpha}.
\end{aligned}\tag{33}$$

Assuming $\zeta^{(i)}$ is nonzero, and taking into account that $\bar{s}^\alpha = -\epsilon^{\alpha\beta} s_\beta^c$, we can decompose $s_\alpha$ as[5]

$$s_\alpha = \epsilon_{\alpha\beta} \bar{c}^{(i)\beta} \zeta^{(i)} + c_\alpha^{(i)} B \zeta^{(i)*},\tag{34}$$

---

[4]For Wilson lines, we focus on the construction of the super-connection since only the WL along a closed curve is truly gauge invariant without subtleties.

[5]In eqs. (34)-(36) and other equations with repeated $(i)$'s, there is no summation with respect to the repeated indices $i$'s.

where $B = \Gamma^{35}$ in our convention.[6] It follows from (32) that

$$u_\alpha = \bar{k}^{(i)} \epsilon_{\alpha\beta} \bar{c}^{(i)\beta} \zeta^{(i)} + k^{(i)} c_\alpha^{(i)} B \zeta^{(i)*}, \tag{35}$$

and

$$\bar{u}^\alpha \Gamma_0 u_\alpha = \bar{k}^{(i)} k^{(i)} \bar{s}^\alpha \Gamma_0 s_\alpha. \tag{36}$$

Then (30) leads to

$$
\begin{aligned}
&\{Q_u G_s, G_s\} = \{Q_s G_s, G_u\} \Longleftrightarrow \{Q_u G_u, G_s\} = K\{Q_s G_s, K G_s\} \\
\Longleftrightarrow\ &(\bar{u}^\alpha \Gamma_0 u_\alpha)\{Q_s G_s, G_s\} = (\bar{s}^\beta \Gamma_0 s_\alpha) k^{(1)} k^{(2)} \{Q_s G_s, G_s\} \\
\Longleftrightarrow\ &k^{(1)} k^{(2)} = \frac{\bar{u}^\alpha \Gamma_0 u_\alpha}{\bar{s}^\alpha \Gamma_0 s_\alpha} \Longleftrightarrow k^{(2)} = \bar{k}^{(1)}.
\end{aligned}
\tag{37}
$$

Repeating the similar analysis for other terms in (32), we find in general $k^{(1)}$ and $k^{(2)}$ are real numbers and the Wilson loop preserves one real Poincaré supercharge. But when $(\zeta^{(1)}, \eta^{(1)}, B\zeta^{(2)*}, B\eta^{(2)*})$ are all proportional to a single spinor, $k^{(1)}$ can be complex. In this case, the Wilson loop preserves two real Poincaré supercharges. In both cases, the Wilson loop cannot preserve any conformal supercharges because $\zeta^c$ and $\eta$ have the dimension of inverse square root of mass. One can check this explicitly by applying the superconformal transformations on $L$. Therefore the Wilson loop is 1/16 or 1/8 BPS with respect to the total 16 supercharges. Compared with the bosonic BPS Wilson lines, the fermionic ones preserve quite fewer supersymmetries. One reason is that all conformal supercharges are broken.

## 2.3 Circular BPS Wilson loops in Euclidean space

In the Euclidean signature, the bars over the spinors do not stand for Dirac conjugation. $\psi$ and $\bar\psi$ are independent spinors. It is convenient to define $\bar{s}^\alpha = -\epsilon^{\alpha\beta} s_\beta^c$ for any spinors with an $\alpha$ index.

Let us start with the 1/2-BPS bosonic connection

$$L_{1/2} = g\dot{x}^m A_m + igr A_5, \tag{38}$$

on the contour of a circle $(x^0, x^1, x^2, x^3) = r(\cos\tau, \sin\tau, 0, 0)$. The supersymmetries preserved by the bosonic Wilson loop $W_{\text{bos}} = \mathcal{P}\exp(i\int_0^{2\pi} d\tau L_{1/2}(\tau))$ satisfy

$$r^{-1}\dot{x}^m \Gamma_m \Gamma_5 \xi_\alpha = i\xi_\alpha \quad \Rightarrow \quad \vartheta_\alpha = -ir^{-1}\Gamma_{015}\theta_\alpha. \tag{39}$$

The dot denotes derivation with respect to $\tau$. We would like to construct a Wilson loop on the same contour which is invariant under a supercharge $\mathcal{Q}_s$ parameterized by

$$\theta_\alpha = \frac{1}{2\sqrt{2}}\theta s_\alpha, \qquad \vartheta_\alpha = -\frac{i}{2\sqrt{2}r}\Gamma_{015}\theta s_\alpha, \tag{40}$$

where $\theta$ is a complex Grassman variable and $s^\alpha$ is a bosonic spinor. On the contour, the supercharge $\mathcal{Q}_s$ acts as:

$$
\begin{aligned}
\mathcal{Q}_s q^\alpha &= -i\bar{s}^\alpha \Pi_- \psi, \\
\mathcal{Q}_s q_\alpha &= i\bar\psi \Pi_+ s_\alpha, \\
\mathcal{Q}_s \psi &= -D_\mu q^\alpha \Gamma^\mu \Pi_+ s_\alpha + ir^{-1} q^\alpha \Gamma_{015} s_\alpha, \\
\mathcal{Q}_s \bar\psi &= \bar{s}^\alpha \Pi_- \Gamma^\mu D_\mu q_\alpha + ir^{-1} \bar{s}^\alpha \Gamma_{015} q_\alpha,
\end{aligned}
\tag{41}
$$

---

[6]Notice that this $B$ is not the one in (14).

where $\Pi_\pm = \frac{1}{2} \pm \frac{i}{2r}\Gamma_5 \dot{x}^m \Gamma_m$. We assume the fermionic part of the super-connection takes the form:

$$F = r\zeta^c \Pi_- \psi + r\bar{\psi}\Pi_+ \eta, \quad \zeta^c = \begin{pmatrix} \zeta^{(1)c} I_N & 0 \\ 0 & \zeta^{(2)c} I_N \end{pmatrix}, \quad \eta = \begin{pmatrix} \eta^{(2)} I_N & 0 \\ 0 & \eta^{(1)} I_N \end{pmatrix}. \quad (42)$$

We find

$$\begin{aligned} \mathcal{Q}_s F =& \zeta^c \Pi_- (-r^{-1} D_\tau q^\alpha \dot{x}^m \Gamma_m s_\alpha - i D_5 q^\alpha \dot{x}^m \Gamma_m s_\alpha + r^{-1} q^\alpha x^m \Gamma_m s_\alpha) \\ & + (r^{-1}\bar{s}^\alpha \dot{x}^m \Gamma_m D_\tau q_\alpha + i\bar{s}^\alpha \dot{x}^m \Gamma_m D_5 q_\alpha - r^{-1}\bar{s}^\alpha x^m \Gamma_m q_\alpha)\Pi_+ \eta. \end{aligned} \quad (43)$$

In order that $\mathcal{Q}_s F$ takes the form of $\partial_\tau G_s + \ldots$, we need

$$\partial_\tau(\zeta^c \Pi_-)\dot{x}^m \Gamma_m s_\alpha = 0, \bar{s}^\alpha \dot{x}^m \Gamma_m \partial_\tau(\Pi_+ \eta) = 0. \quad (44)$$

As discussed in appendix B.1, if $s_1$ and $s_2$ are linearly dependent, the Wilson loop is BPS only when $\mathcal{Q}_s L = 0$. In Lorentzian signature, linear dependence of $s_1$ and $s_2$ is not consistent with the reality condition. In the following of this section, we assume that $s_1$ and $s_2$ are linearly independent.

Solving the differential equations (44) for $\zeta^c \Pi_-$ and $\Pi_+ \eta$, we find the general solutions can be represented by $\tau$-independent $\zeta^c$ and $\eta$ which satisfy

$$\zeta^c \Gamma_{015} s_\alpha = \bar{s}^\alpha \Gamma_{015} \eta = 0. \quad (45)$$

Then we get

$$G_s = i\zeta^c \Pi_- \Gamma_5 s_\alpha q^\alpha - iq_\alpha \bar{s}^\alpha \Gamma_5 \Pi_+ \eta. \quad (46)$$

Acting $\mathcal{Q}_s$ on $G_s$, we find

$$\begin{aligned} \mathcal{Q}_s G_s &= \zeta^c \Pi_- \Gamma_5 s_\alpha(\bar{s}^\alpha \Pi_- \psi) + \bar{s}^\alpha \Gamma_5 \Pi_+ \eta(\bar{\psi}\Pi_+ s_\alpha) \\ &= \frac{1}{2}\bar{s}^\alpha \Pi_- \Gamma_5 s_\alpha(\zeta^c \Pi_- \psi) + \frac{1}{2}\bar{s}^\alpha \Gamma_5 \Pi_+ s_\alpha(\bar{\psi}\Pi_+ \eta) \\ &= \frac{1}{2r}\bar{s}^\alpha \Pi_- \Gamma_5 s_\alpha F. \end{aligned} \quad (47)$$

Similar to the straight line case, one can obtain $B$ from the conditions $[B, G_s] = 0$ and $Q_s B = i\{F, G_s\}$. The result is,

$$B = i\frac{2r}{\bar{s}^\alpha \Pi_- \Gamma_5 s_\alpha}G_s^2. \quad (48)$$

Finally the super-connection $L$ is

$$L = L_{1/2} + \frac{2r}{\bar{s}^\alpha \Pi_- \Gamma_5 s_\alpha}\mathcal{Q}_s G_s + i\frac{2r}{\bar{s}^\alpha \Pi_- \Gamma_5 s_\alpha}G_s^2, \quad (49)$$

which satisfies $\mathcal{Q}_s L = \mathcal{D}_\tau G_s$. Because $G_s$ is periodic on the contour, the trace of the holonomy of $L$ does not preserve the supercharge $\mathcal{Q}_s$, which is different from their three-dimensional counterparts [27]. Since $L$ has a natural supermatrix structure, we can define the Wilson loop by using the supertrace:

$$W_{\text{fer}} = s\text{Tr}\mathcal{P}\exp\left(i\oint L d\tau\right), \quad (50)$$

which preserves the supercharge $\mathcal{Q}_s$. Following similar steps as in the three-dimensional case [27,45], one can show that the condition $\mathcal{Q}_s L = \mathcal{D}_\tau G_s$ leads to a classical $\mathcal{Q}_s$-cohomological equivalence between the fermionic BPS Wilson loop and the bosonic one:

$$W_{\text{fer}} - W_{\text{bos}} = \mathcal{Q}_s V, \quad (51)$$

where

$$W_{\text{bos}} = \text{sTr}\mathcal{P}\exp\left(i\oint L_{1/2}d\tau\right), \tag{52}$$

and $V$ is a complicated function of the gauge and matter fields whose first few orders are

$$V = \text{sTr}\mathcal{P}\left(e^{i\oint L_{1/2}d\tau}\left(\oint \Lambda(\tau_1)d\tau_1 + \frac{i}{2}(\Lambda * F - F * \Lambda)\right.\right.$$
$$\left.\left. + i\Lambda * B - iB * \Lambda - \Lambda * F * F - F * \Lambda * F - F * F * \Lambda + \dots\right)\right), \tag{53}$$

where $\Lambda = 2irG_s/(\bar{s}^\alpha\Pi_-\Gamma_5 s_\alpha)$ we use the notation

$$X * Y = \int_{\tau_1 > \tau_2} d\tau_1 d\tau_2 X(\tau_1)Y(\tau_2), \tag{54}$$

$$X * Y * Z = (X * Y) * Z. \tag{55}$$

The complete construction of $V$ can be performed following the procedure in appendix D of [45].

To investigate the possible supersymmetry enhancements of the Wilson loop, we need to solve all the $u_\alpha$ satisfying[7]

$$\mathcal{Q}_u L = \mathcal{D}_\tau G_u, \tag{56}$$

where $G_u = i\zeta^c\Pi_-\Gamma_5 u_\alpha q^\alpha - iq_\alpha \bar{u}^\alpha\Gamma_5\Pi_+\eta$. Detailed discussion of the solutions are relegated to appendix B.2. Here we summarize the conclusion as follows:

- When $\zeta^{(i)}$ and $\eta^{(i)}$ are all proportional to a nonzero bosonic spinor $\chi$ satisfying $\chi^c\Gamma_{015}\chi = 0$, the Wilson loop is 3/16-BPS. Denoting the two-dimensional vector space spanned by $\{\zeta^{(1)}, \Gamma_{01}\zeta^{(1)}, \Gamma_{15}\zeta^{(1)}, \Gamma_{50}\zeta^{(1)}\}$ by $V_{\zeta^{(1)}}$, $s_\alpha$ can be decomposed as

$$s_\alpha = c_\alpha s_{\perp\zeta^{(1)}} + s_{\alpha\|\zeta^{(1)}}, \tag{57}$$

  where $s_{\alpha\|\zeta^{(1)}}$ is a vector in $V_{\zeta^{(1)}}$ and $s_{\perp\zeta^{(1)}}$ is a vector in the complementary transverse direction. The preserved supercharges can be parameterized by

$$u_\alpha = k^{(1)}s_\alpha + c_\alpha u'_{\|\zeta^{(1)}}, \tag{58}$$

  where $u'_{\|\zeta^{(1)}}$ is an arbitrary vector in $V_{\zeta^{(1)}}$. There are three complex parameters, one in $k^{(1)}$ and two in $u'_{\|\zeta^{(1)}}$, so the Wilson loop is 3/16-BPS. In another word, taking $u'_{\|\zeta^{(1)},1}$ and $u'_{\|\zeta^{(1)},2}$ to be one basis of $V_{\zeta^{(1)}}$, $u_\alpha$ is in the three-dimensional complex space spanned by $s_\alpha$, $c_\alpha u'_{\|\zeta^{(1)},1}$ $c_\alpha u'_{\|\zeta^{(1)},2}$ for each $\alpha$.

- When $\zeta^{(1)} \propto \eta^{(1)}$, $\zeta^{(2)} \propto \eta^{(2)}$, $\zeta^{(i)c}\Gamma_{015}\zeta^{(i)} = 0$ and $V_{\zeta^{(1)}} \cap V_{\zeta^{(2)}} = \{0\}$. The Wilson loop is 1/8-BPS. The preserved supercharges can be parameterized by

$$u_\alpha = k^{(2)}c_\alpha^{(1)}s_{\|\zeta^{(1)}} + k^{(1)}c_\alpha^{(2)}s_{\|\zeta^{(2)}}. \tag{59}$$

  There are two complex parameters $k^{(1)}$ and $k^{(2)}$, so the Wilson loop is 1/8-BPS. Now $u_\alpha$ is in the two-dimensional complex space spanned by $c_\alpha^{(1)}s_{\|\zeta^{(1)}}$ and $c_\alpha^{(2)}s_{\|\zeta^{(2)}}$ for each $\alpha$.

---

[7]More precisely speaking, we should exclude other types of combinations of Poincarè superchanges and conformal superchanges. We conjecture that the combinations other than $Q_u$ will not be preserved by the above fermionic WLs.

- When $\zeta^{(i)}$ and $\eta^{(i)}$ are all proportional to a nonzero bosonic spinor $\chi$ satisfying $\chi^c \Gamma_{015} \chi \neq 0$ and

$$(s_1^c \Gamma_{015} s_1)(s_2^c \Gamma_{015} s_2) - (s_1^c \Gamma_{015} s_2)^2 = 0, \tag{60}$$

the Wilson loop is 1/8-BPS. In this case it is not easy to list a basis of the linear space which $u_\alpha$'s belong to. We refer to the appendix B.2 for details.

- Otherwise, the Wilson loop is 1/16-BPS. $u_\alpha$ is in the one-dimensional complex space spanned by $s_\alpha$ for each $\alpha$. In another word, all preserved supercharges are proportional to $\mathcal{Q}_s$.

# 3 Fermionic BPS Wilson loops in $\mathcal{N} = 4$ SYM

Since the $\mathcal{N} = 2$ theory considered in the previous section can be obtained from the $\mathcal{N} = 4$ SYM by orbifolding followed by a marginal deformation, one might expect to find fermionic BPS Wilson loops in $\mathcal{N} = 4$ SYM. In this section, we turn to the $\mathcal{N} = 4$ SYM and construct the fermionic BPS Wilson loops. As in the $\mathcal{N} = 2$ case, line BPS Wilson loops in Minkowski spacetime in general preserve one real supercharge and circular BPS Wilson loops in Euclidean space preserve one complex supercharge. We also give some examples of fermionic Wilson loops preserving more supercharges.

## 3.1 BPS Wilson lines in Minkowski spacetime

The action of $\mathcal{N} = 4$ SYM is

$$S_{\mathcal{N}=4} = \int_{\mathbf{R}^4} d^4 x \left( -\frac{1}{4} \mathrm{Tr}(F_{MN} F^{MN}) - \frac{i}{2} \mathrm{Tr}(\bar{\Psi} \Gamma^M D_M \Psi) \right). \tag{61}$$

In this section $\Gamma^M$'s are 10d gamma matrices. We use the index conventions $M, N = 0, \ldots, 9$ and $R, S = 5, \ldots, 9$. The action is invariant under the superconformal transformations:

$$\begin{aligned}
\delta A_M &= -i \xi^c \Gamma_M \Psi, \\
\delta \Psi &= \frac{1}{2} F_{MN} \Gamma^{MN} \xi - 2 \Gamma^S A_S \vartheta,
\end{aligned} \tag{62}$$

where $\xi = \theta + x^m \Gamma_m \vartheta$ with $m = 0, \ldots, 3$. The constant spinors $\theta$ and $\vartheta$ generate Poincaré supersymmetry transformations and special superconformal transformations respectively.

The supersymmetries preserved by the bosonic 1/2-BPS connection [7,8]

$$L_{1/2} = g A_0 - g A_5, \tag{63}$$

on the straight line $x^m = \delta_0^m \tau$ satisfy

$$\Gamma_5 \Gamma_0 \xi = \xi. \tag{64}$$

In this subsection we focus on Poincaré supersymmetries and set $\vartheta = 0$. We would like to construct a fermionic Wilson loop with a connection

$$L = L_{1/2} + B + F, \tag{65}$$

on the same contour which is invariant under a supercharge $Q_s$ where $s$ is a bosonic spinor satisfying $\Gamma_5 \Gamma_0 s = s$. On the contour, the supercharge $Q_s$ acts as:

$$\begin{aligned}
Q_s A_M &= -i \bar{s} \Gamma_M \Psi, \\
Q_s \Psi &= \frac{1}{2} F_{MN} \Gamma^{MN} s.
\end{aligned} \tag{66}$$

We require $Q_s L = \mathcal{D}_0 G_s$ and assume the bosonic matrix $G_s$ takes the form of $G_s = m^S A_S$ where $m^S$ is a vector in the directions $4, \ldots, 9$. Acting $Q_s$ twice on $G_s$, we find

$$Q_s G_s = -i m^S \bar{s} \Gamma_S \Psi, \tag{67}$$

$$Q_s^2 G_s = -\frac{i}{2} m^S \bar{s} \Gamma_S F_{MN} \Gamma^{MN} s = -i F_{SN} \bar{s} m^S \Gamma^N s = -i(\bar{s}\Gamma^5 s)(\partial_0 G_s - i[L_{1/2}, G_s]). \tag{68}$$

Then a connection $L$ satisfying $Q_s L = \mathcal{D}_0 G_s$ is

$$L = L_{1/2} + \frac{i}{\bar{s}\Gamma^5 s} Q_s G_s - \frac{1}{\bar{s}\Gamma^5 s} G_s^2, \tag{69}$$

and the Wilson line $W_{\text{fer}} = \mathcal{P} \exp\left(i \int L d\tau\right)$ preserves the supercharge $Q_s$. More generally, we can take $G_s = M^S \otimes A_S$ where $M^S$ is an $r \times r$ matrix-valued vector and the connection becomes

$$L = I_r \otimes L_{1/2} + \frac{i}{\bar{s}\Gamma^5 s} M^S \otimes Q_s A_S - \frac{1}{\bar{s}\Gamma^5 s} (M^S \otimes A_S)^2. \tag{70}$$

We now consider supersymmetry enhancement. Acting another supercharge $Q_u$ with the condition $\Gamma_5 \Gamma_0 u = u$ on $F$, we find

$$Q_u F = \frac{1}{2(\bar{s}\Gamma^5 s)} M^S \otimes F_{MN} \bar{s} \Gamma_S \Gamma^{MN} u. \tag{71}$$

In order that $Q_u F$ takes the form of $\partial_\tau G_u + \ldots$, the terms with $M, N = 1, 2, 3$ must vanish. One way is to take

$$M^4 = M^5 = 0, \qquad \Gamma^{6789} s = -s, \qquad \Gamma^{6789} u = -u. \tag{72}$$

Then we have

$$\begin{aligned} Q_u F &= \frac{1}{(\bar{s}\Gamma^5 s)} M^P \bar{s} \Gamma_P \Gamma^{Q5} u \otimes (F_{0Q} - F_{5Q}) \\ &= \partial_0 G_u - i[I_r \otimes L_{1/2}, G_u], \end{aligned} \tag{73}$$

with

$$G_u = M^P U_P{}^Q \otimes A_Q, \qquad U_P{}^Q \equiv \frac{\bar{s}\Gamma_P \Gamma^{Q5} u}{\bar{s}\Gamma^5 s}. \tag{74}$$

We use the letters $P$ and $Q$ in the range of $6, \ldots, 9$. Using the identity,

$$\frac{1}{\bar{u}\Gamma^5 u} (\bar{s}\Gamma_P \Gamma^{Q5} u) \bar{u} \Gamma_Q = \bar{s} \Gamma_P, \tag{75}$$

we find

$$F = \frac{i}{\bar{s}\Gamma^5 s} Q_s G_s = \frac{i}{\bar{u}\Gamma^5 u} Q_u G_u, \tag{76}$$

and therefore $Q_u F = \partial_\tau G_u - i[L_{1/2}, G_u]$ is satisfied.

By using an explicit representation of the $\Gamma$-matrices, one can show that the matrix $U^T$ has two distinct eigenvalues $\lambda_+$ and $\lambda_-$, which satisfy

$$\lambda_+ \lambda_- = \frac{\bar{u}\Gamma^5 u}{\bar{s}\Gamma^5 s}, \tag{77}$$

and each eigenvalue corresponds to a two-dimensional eigenspace. So we can write $G_u$ as

$$G_u = \lambda_+ M_+^P \otimes A_P + \lambda_- M_-^P \otimes A_P, \quad M^P = M_+^P + M_-^P, \quad M_\pm^P U_P{}^Q = \lambda_\pm M_\pm^Q, \tag{78}$$

and we have

$$G_u^2 = (\lambda_+^2 M_+^P M_+^Q + \lambda_-^2 M_-^P M_-^Q + \frac{\bar{u}\Gamma^5 u}{\bar{s}\Gamma^5 s} M_+^P M_-^Q + \frac{\bar{u}\Gamma^5 u}{\bar{s}\Gamma^5 s} M_-^P M_+^Q) \otimes A_P A_Q. \tag{79}$$

Therefore if

$$M_+^P M_+^Q = M_-^P M_-^Q = 0, \tag{80}$$

we have

$$\frac{1}{\bar{u}\Gamma^5 u} G_u^2 = \frac{1}{\bar{s}\Gamma^5 s} G_s^2. \tag{81}$$

and the conditions

$$[G_s^2, G_u] = 0, \{Q_u G_s, G_s\} = \{Q_s G_s, G_u\} \tag{82}$$

are satisfied. To summarize, the Wilson line preserves the supercharge $Q_u$ when (72) and (80) are satisfied. In the following, we provide a simple example to illustrate our construction. Let the Wilson line preserve supercharge $Q_s$ with $s$ satisfying

$$\Gamma_{50}s = -\Gamma_{6789}s = -\Gamma_{2367}s = s. \tag{83}$$

The solutions to these constraints are linear combinations of two linearly independent Majorana-Weyl spinors. By using an explicit representation of the $\Gamma$-matrices, one can show that the matrix $U$ has the form of

$$U_P{}^Q = \begin{pmatrix} p_1 & p_2 & 0 & 0 \\ -p_2 & p_1 & 0 & 0 \\ 0 & 0 & p_1 & p_2 \\ 0 & 0 & -p_2 & p_1 \end{pmatrix}, \tag{84}$$

where $p_1$ and $p_2$ are constants depending on the $u$ and $s$ we choose. The eigenvectors of $U^T$ are

$$\begin{array}{llll} \text{eigenvalue} & p_1 + ip_2: & m_{1+} = (0,0,i,1), & m_{2+} = (i,1,0,0), \\ \text{eigenvalue} & p_1 - ip_2: & m_{2-} = (0,0,-i,1), & m_{2-} = (-i,1,0,0). \end{array} \tag{85}$$

Therefore we can take

$$M^P = K^i m_{i-}^P + J^i m_{i+}^P, \tag{86}$$

where the matrices $K^i$ and $J^i$ satisfies

$$K^i K^j = J^i J^j = 0. \tag{87}$$

## 3.2 Circular BPS Wilson loops in Euclidean space

In the Euclidean signature, the superconformal transformations are formally the same as (62), but there are no reality conditions for the spinors. The supersymmetries preserved by the bosonic 1/2-BPS connection [7,8]

$$L_{1/2} = g\dot{x}^\mu A_\mu + igr A_5, \tag{88}$$

on the circular contour $(x^0, x^1, x^2, x^3) = r(\cos\tau, \sin\tau, 0, 0)$ satisfy

$$\dot{x}^\mu \Gamma_\mu \Gamma_5 \xi = i\xi \quad \Rightarrow \quad \vartheta = -ir^{-1}\Gamma_{015}\theta. \tag{89}$$

We would like to construct a Wilson loop on the same contour which is invariant under a supercharge $\mathcal{Q}_s$ parameterized by

$$\theta = \frac{1}{2}\chi s, \qquad \vartheta = -\frac{i}{2r}\Gamma_{015}\chi s, \tag{90}$$

where $\chi$ is a complex Grassmann variable and $s$ is a bosonic spinor. On the contour, the supercharge $\mathcal{Q}_s$ acts as:

$$\mathcal{Q}_s A_M = -is^c \Pi_- \Gamma_M \Psi,$$
$$\mathcal{Q}_s \Psi = \frac{1}{2} F_{MN} \Gamma^{MN} \Pi_+ s + ir^{-1} \Gamma^S A_S \Gamma_{015} s, \tag{91}$$

where $\Pi_\pm = \frac{1}{2} \pm \frac{i}{2r} \Gamma_5 \dot{x}^m \Gamma_m$. The connection $L = L_{1/2} + B + F$ is expected to transform as $\mathcal{Q}_s L = \mathcal{D}_\tau G_s$. We assume the bosonic matrix $G_s$ take the form of $G_s = m^S A_S$ where $m^S$ is a vector in the directions $4, \ldots, 9$. Acting $\mathcal{Q}_s$ twice on $G_s$, we find

$$\begin{aligned}
\mathcal{Q}_s^2 G_s &= -\frac{i}{2} m^R s^c \Pi_- \Gamma_R F_{MN} \Gamma^{MN} \Pi_+ s + r^{-1} m^R s^c \Pi_- \Gamma_R \Gamma^S A_S \Gamma_{015} s \\
&= -iF_{RN} s^c \Pi_- m^R \Gamma^N \Pi_+ s + r^{-1} m^R s^c \Pi_- \Gamma_R \Gamma^S A_S \Gamma_{015} s \\
&= im^R F_{5R} s^c \Pi_- \Gamma^5 s + r^{-1} m^R \dot{x}^m F_{mR} s^c \Pi_- \Gamma^5 s + r^{-1} m^R s^c \Pi_- \Gamma_R \Gamma^S A_S \Gamma_{015} s \\
&= r^{-1} (s^c \Pi_- \Gamma^5 s)(\dot{x}^m \partial_m G_s - i[L_{1/2}, G_s]) \\
&\quad - r^{-1} \dot{m}^R A_R s^c \Pi_- \Gamma^5 s - im^R A_S \frac{s^c x^m \Gamma_m \delta_R^S + irs^c \Gamma_{015} \Gamma_R \Gamma^S s}{2r^2}.
\end{aligned} \tag{92}$$

Therefore we require

$$\dot{m}^S + im^R \frac{r^{-1} s^c x^m \Gamma_m s \delta_R^S + is^c \Gamma_{015} \Gamma_R \Gamma^S s}{2s^c \Pi_- \Gamma^5 s} = 0. \tag{93}$$

The solution is

$$m^S(\tau) = c^R s^c \Pi_- \Gamma_5 s (\exp M)_R^S, \tag{94}$$

where the matrix $M$ is defined by

$$M_R^S = \int^\tau d\tau' \frac{s^c \Gamma_{015} \Gamma_R \Gamma^S s}{2s^c \Pi_-(\tau') \Gamma^5 s}. \tag{95}$$

It is not hard to obtain that

$$m^S(\tau) = c^R s^c \Pi_- \Gamma_5 s \left[ \exp \left( -\frac{2iM_{015}}{\sqrt{v_0^2 + v_1^2 + v_5^2}} \tanh^{-1} \left( \frac{v_0 + (v_1 + iv_5) \tan\left(\frac{\tau}{2}\right)}{\sqrt{v_0^2 + v_1^2 + v_5^2}} \right) \right) \right]_R^S, \tag{96}$$

where $v_\mu = s^c \Gamma_\mu s$ and $(M_{015})_R^S = s^c \Gamma_{015} \Gamma_R \Gamma^S s$. The matrix $M_{015}$ satisfies

$$M_{015}^5 - \frac{1}{2} \text{Tr}(M_{015}^2) M_{015}^3 - (\frac{1}{2} \text{Tr}(M_{015}^2) + v^2) v^2 M_{015} = 0, \tag{97}$$

where $v^2 = v_0^2 + v_1^2 + v_5^2$. Therefore the exponential $\exp M$ takes the form:

$$C_1 e^{f(\tau)\sqrt{-1 - \frac{\text{Tr} M_{015}^2}{2v^2}}} + C_2 e^{-f(\tau)\sqrt{-1 - \frac{\text{Tr} M_{015}^2}{2v^2}}} + C_3 e^{f(\tau)} + C_4 e^{-f(\tau)} + C_5, \tag{98}$$

where $C_i$ are $\tau$ independent matrices and

$$f(\tau) = 2 \tanh^{-1} \left( \frac{v_0 + (v_1 + iv_5) \tan\left(\frac{\tau}{2}\right)}{\sqrt{v_0^2 + v_1^2 + v_5^2}} \right). \tag{99}$$

From

$$e^{f(\tau)} = \frac{e^{i\tau}(v + v_0 - iv_1 + v_5) + v + v_0 + iv_1 - v_5}{e^{i\tau}(v - v_0 + iv_1 - v_5) + v - v_0 - iv_1 + v_5},$$ (100)

we know that $e^{f(\tau)}$ is periodic with period $2\pi$, then the first two terms in (98) are periodic when

$$\sqrt{-1 - \frac{\mathrm{Tr} M_{015}^2}{2v^2}} \in \mathbb{Z}.$$ (101)

A simple example is when $\Gamma^{6789}s = -s$, we have

$$\sqrt{-1 - \frac{\mathrm{Tr} M_{015}^2}{2v^2}} = 1.$$ (102)

As far as we know, it is for the first time that we need to impose extra constraints to guarantee the (anti-)periodicity of $G_s$ in the construction of circular fermionic BPS WLs.

A connection $L$ which satisfies $\mathcal{Q}_s L = \mathcal{D}_\tau G_s$ is

$$L = L_{1/2} + \frac{r}{s^c \Pi_- \Gamma^5 s} \mathcal{Q}_s G_s + \frac{ir}{s^c \Pi_- \Gamma^5 s} G_s^2.$$ (103)

One can generalize $m^S$ to an $r \times r$ matrix-valued vector $M^S$ by taking $c^R$ in (96) to be a $r$-dimensional constant matrix and the connection becomes

$$L = I_r \otimes L_{1/2} + \frac{r}{s^c \Pi_- \Gamma^5 s} M^S \otimes \mathcal{Q}_s A_S + \frac{ir}{s^c \Pi_- \Gamma^5 s} (M^S \otimes A_S)^2.$$ (104)

Because $G_s$ is periodic on the contour provided (101) is satisfied, to construct a BPS Wilson loop we need $L$ to be a supermatrix and only off-diagonal blocks of $M^S$ are nonzero. Explicitly, we demand $M^S$ to be

$$M^S = \begin{pmatrix} 0 & M_1^S \\ M_2^S & 0 \end{pmatrix}.$$ (105)

And then $L$ can be decomposed as

$$L = \begin{pmatrix} B_1 & F_1 \\ F_2 & B_2 \end{pmatrix}.$$ (106)

Then a BPS Wilson loop preserving the supercharge $\mathcal{Q}_s$ can be defined as

$$W_{\mathrm{fer}} = s\mathrm{Tr}\mathcal{P} \exp\left( i \oint L d\tau \right).$$ (107)

As the case in the previous section, one can prove that $W_{\mathrm{fer}} - W_{\mathrm{bos}} = \mathcal{Q}_s V$ where

$$W_{\mathrm{bos}} = s\mathrm{Tr}\mathcal{P} \exp\left( i \oint (I_r \otimes L_{1/2}) d\tau \right),$$ (108)

with sTr defined as the one in (107), and $V$ takes a similar form as that given by (53) with now $\Lambda = ir G_s / (s^c \Pi_- \Gamma^5 s)$.

It would be difficult to find all possible supersymmetry enhancements. Leaving it for future investigation, here we give a simple example of a Wilson loop preserving two supercharges $\mathcal{Q}_s$ and $\mathcal{Q}_u$ with $s$ and $u$ satisfying

$$i\Gamma^{01}s = -\Gamma^{6789}s = s, \quad \Gamma^{6789}u = -u, \quad u = \Gamma^{67}s.$$ (109)

If we take

$$M^4 = M^5 = 0, \quad M^P = K^i m_{i-}^P + J^i m_{i+}^P,$$ (110)

where the vectors $m_{i\pm}$ are given in (85) and the matrices $K^i$ and $J^i$ satisfy $K^i K^j = J^i J^j = 0$, the Wilson loop will be invariant under $\mathcal{Q}_s$ and $\mathcal{Q}_u$.

# 4  Conclusion and discussions

In this paper we have constructed fermionic BPS Wilson loops in $\mathcal{N} = 2$ quiver theory with gauge group $SU(N) \times SU(N)$ and $\mathcal{N} = 4$ SYM. The connections of these fermionic BPS Wilson have a supermatrix structure. We have constructed BPS Wilson lines in Minkowski spacetime and BPS circular Wilson loops in Euclidean space.

There are at least two features of such four-dimensional fermionic WLs which are quite different from the such WLs in ABJM theory: the first is that the fermionic WLs along straight lines in four-dimensional break the scale invariance, and the second is that the fermionic WLs in four dimensions preserve a small part of the supercharges preserving by the bosonic WLs. The common feature of four-dimensional WLs and three-dimensional WLs in [27–29] is that the fermionic WL is always in the same $Q$-cohomology class of a bosonic WL with $Q$ being a supercharge shared by these two WLs. This was explicitly illuminated at the classical level. Assuming that this is also correct at the quantum level, we predict that the fermionic BPS WL has the same vev as one of $W_{\text{bos}}$. For circular loops, the vevs of $W_{\text{bos}}$ defined using supertrace can be obtained from the results in [13,42,46,47]. It is valuable to check this prediction from $Q$-cohomology at the quantum level through direct perturbative computations, as people have already done in three-dimensional cases [48–50]. Comparing vevs of WLs obtained from perturbative computations and localization in three-dimensional super-Chern-Simons theories is subtle because they depend on the choice of framing which is a result of the point-splitting regularization prescription of the perturbative expansion of a WL. It was suggested [51] that one should choose framing $-1$ for perturbative computations of vevs of bosonic WLs to compare with the prediction from localization. And in [52], a suitable regularization scheme within this framing for fermionic BPS WLs was proposed. But the computations in this scheme are quite complicated in practice. We expect that comparing WL vevs in four-dimensional SYM would be easier because of the absence of framing dependence in four dimensions.

In our construction of fermionic WLs, we started with half-BPS bosonic WLs along a line or circle. Since there are various bosonic WLs with fewer supersymmetries, it is interesting to construct BPS fermionic WLs starting with these loops. One may first start with 1/4-BPS Wilson loops along a latitude circle [20, 21, 53], since they are almost the simplest among WLs with fewer supersymmetries. Another way to include fermions inside the WLs is based on $\mathcal{N} = 4$ non-chiral superspace [39].[8] One big difference is that the WLs there are along contours in superspace instead of ordinary spacetime. The possible relation between the loops there and the ones constructed here certainly deserves investigation.

The bosonic WLs in $\mathcal{N} = 4$ SYM are dual to (Wilson-)'t Hooft loops under S-duality transformation [5, 54] and F-stings/D-branes/bubbling geometries [7, 8, 55–61] under AdS/CFT correspondence. It is interesting but also challenging to study the S-dual and the holographic dual of the fermionic WLs constructed here.

Bosonic WLs play at least two roles in the study of integrability of the planar $\mathcal{N} = 4$ SYM theory. When we insert composite local operators into the WLs, WLs provide integrable boundary conditions/interactions for the open spin chains from the composite operators [62, 63].[9] When we consider the correlators of a half-BPS circular WL (in the fundamental or an antisymmetric representation) and a non-BPS single trace operator in the 't Hooft limit, this WL will provide an integrable matrix product state [64]. It is appealing to explore whether the fermionic WLs constructed here also have such an integrable structure.

The construction of fermionic WLs involves dimensionful parameters which lead to the breaking of scale invariance. So such WLs will lead to defect quantum field theories (dQFTs),

---

[8]These WLs were studied in [40] using integral forms. See also an old related construction in [41] and the construction of the supersymmetrized WLs in appendix C of [9].

[9]The usual Wilson loops also provide integrable boundary conditions/interactions [63].

instead of defect conformal field theories (dCFTs). On the other hand, by taking fermionic WLs as deformations of bosonic BPS WLs suggested by the construction, these WLs lead to irrelevant deformations of dCFTs. It is interesting to study possible ultraviolet completion of such irrelevant deformations.

## Acknowledgments

We would like to thank Bin Chen, Nadav Drukker, Yunfeng Jiang, Pujian Mao, Takao Suyama, Jia Tian, Yifan Wang, Yi-Nan Wang, Konstantin Zarembo and Jiaju Zhang for very helpful discussions. Special thanks to the anonymous referees of SciPost Physics for very helpful suggestions on improving the manuscript. The work of JW is supported by the National Natural Science Foundation of China, Grant No. 11975164, 11935009, 12047502, and Natural Science Foundation of Tianjin under Grant No. 20JCYBJC00910. The work of HO is supported by the National Natural Science Foundation of China, Grant No. 12205115.

## A  Conventions

In Lorentzian signature, we use the following representation for the 10d gamma matrices:

$$
\begin{aligned}
\Gamma^{\mu}_{(10)} &= I_4 \otimes \Gamma^{\mu}_{(6)}, \quad \mu = 0, \ldots, 5, \\
\Gamma^{s}_{(10)} &= \Gamma^{10-s}_{(4)} \otimes \Gamma^{012345}_{(6)}, \quad s = 6, \ldots, 9.
\end{aligned}
\tag{A.1}
$$

The 4d gamma matrices in Euclidean signature are defined as

$$
\Gamma^{j}_{(4)} = \begin{pmatrix} 0 & -i\sigma_j \\ i\sigma_j & 0 \end{pmatrix}, \quad
\Gamma^{4}_{(4)} = \begin{pmatrix} 0 & I_2 \\ I_2 & 0 \end{pmatrix},
\tag{A.2}
$$

and the 6d gamma matrices in Lorentzian signature are defined as

$$
\begin{aligned}
\Gamma^{0}_{(6)} &= (i\sigma_2) \otimes (-\sigma_3) \otimes (-\sigma_3), \\
\Gamma^{1}_{(6)} &= (\sigma_1) \otimes (-\sigma_3) \otimes (-\sigma_3), \\
\Gamma^{2}_{(6)} &= I_2 \otimes \sigma_1 \otimes (-\sigma_3), \\
\Gamma^{3}_{(6)} &= I_2 \otimes \sigma_2 \otimes (-\sigma_3), \\
\Gamma^{4}_{(6)} &= I_2 \otimes I_2 \otimes \sigma_1, \\
\Gamma^{5}_{(6)} &= I_2 \otimes I_2 \otimes \sigma_2,
\end{aligned}
\tag{A.3}
$$

where $\sigma_j$'s are Pauli matrices. The charge conjugate matrices are defined as

$$
C_{(10)} = C_{(4)} \otimes C_{(6)},
\tag{A.4}
$$

$$
C_{(4)} = -\Gamma^{13}_{(4)} = \begin{pmatrix} i\sigma_2 & 0 \\ 0 & i\sigma_2 \end{pmatrix},
\tag{A.5}
$$

$$
C_{(6)} = \Gamma^{035}_{(6)} = \begin{pmatrix}
0 & 0 & 0 & 0 & 0 & 0 & 0 & 1 \\
0 & 0 & 0 & 0 & 0 & 0 & -1 & 0 \\
0 & 0 & 0 & 0 & 0 & 1 & 0 & 0 \\
0 & 0 & 0 & 0 & -1 & 0 & 0 & 0 \\
0 & 0 & 0 & -1 & 0 & 0 & 0 & 0 \\
0 & 0 & 1 & 0 & 0 & 0 & 0 & 0 \\
0 & -1 & 0 & 0 & 0 & 0 & 0 & 0 \\
1 & 0 & 0 & 0 & 0 & 0 & 0 & 0
\end{pmatrix},
\tag{A.6}
$$

and the charge conjugation of a spinor $\xi$ is defined as $\xi^c \equiv \xi^T C_{(10)}$. The supersymmetry transformation parameter in the $\mathcal{N} = 4$ satisfies the the chirality condition $\Gamma^{0123456789}_{(10)}\xi = \xi$ and the reality condition $\bar{\xi} = \xi^c$.

To derive the $\mathcal{N} = 2$ supersymmetry transformation, we write the $\mathcal{N} = 4$ supersymmetry transformation parameters as

$$\xi = \begin{pmatrix} \xi^{\dot{1}} \\ \xi^{\dot{2}} \\ \xi_1 \\ \xi_2 \end{pmatrix}, \tag{A.7}$$

where the components are 6d spinors. The chirality condition can be written as

$$\Gamma^{012345}_{(6)}(\xi^{\dot{1}}, \xi^{\dot{2}}, \xi_1, \xi_2) = (\xi^{\dot{1}}, \xi^{\dot{2}}, -\xi_1, -\xi_2), \tag{A.8}$$

and the reality condition can be written as

$$(\bar{\xi}_{\dot{1}}, \bar{\xi}_{\dot{2}}, \bar{\xi}^1, \bar{\xi}^2) = (-\xi^{\dot{2}}, \xi^{\dot{1}}, -\xi_2, \xi_1)^T C_{(6)} \equiv (-\xi^{\dot{2}c}, \xi^{\dot{1}c}, -\xi_2^c, \xi_1^c). \tag{A.9}$$

The $\mathcal{N} = 2$ supersymmetry transformation parameters satisfies $\Gamma^{6789}_{(10)}\xi = -\xi$:

$$\xi = \begin{pmatrix} 0 \\ 0 \\ \xi_1 \\ \xi_2 \end{pmatrix}. \tag{A.10}$$

The fermionic fields in the $\mathcal{N} = 2$ theory can be reduced from $\Psi$ in the $\mathcal{N} = 4$ theory using

$$\Psi = \begin{pmatrix} \psi \\ -C_{(6)}\bar{\psi}^T \\ \lambda_1 \\ \lambda_2 \end{pmatrix}. \tag{A.11}$$

In Euclidean signature, we use $\Gamma^0_{E(10,6)} = i\Gamma^0_{(10,6)}$. In subsections 2.3 and 3.2, we use the Euclidean Gamma matrices and omit the superscript $E$. We use the same definition of the charge conjugate matrix $C$ in both signatures. There are no reality conditions for the supersymmetry transformation parameters in Euclidean signature.

## B  Technical details for section 2.3

### B.1  The case when $s_1 \propto s_2$

When $s_1 \propto s_2$, the solutions to (44) are more complicated. Assuming (44) is satisfied, $G_s$ is still given by (46) but

$$\mathcal{Q}_s G_s = \frac{1}{2r}\bar{s}^\alpha \Pi_- \Gamma_5 s_\alpha F = 0. \tag{B.1}$$

The general form of $B$ is

$$B = R^{\alpha\beta}q_\alpha q_\beta + R_\alpha^{\ \beta}q^\alpha q_\beta + R_{\alpha\beta}q^\alpha q^\beta + R^\alpha_{\ \beta}q_\alpha q^\beta. \tag{B.2}$$

Then $\mathcal{Q}_s B = ig\{F, G_s\}$ gives

$$R^{\alpha\beta}\mathcal{Q}_s q_\alpha q_\beta + R_\alpha^{\ \beta}\mathcal{Q}_s q^\alpha q_\beta + R_{\alpha\beta}\mathcal{Q}_s q^\alpha q^\beta + R^\alpha_{\ \beta}\mathcal{Q}_s q_\alpha q^\beta \tag{B.3}$$

$$= (\zeta^c\Pi_-\psi + \bar{\psi}\Pi_+\eta)(i\zeta^c\Pi_-\Gamma_5 s_\alpha q^\alpha - iq_\alpha \bar{s}^\alpha\Gamma_5\Pi_+\eta)$$

$$\Rightarrow R^{\alpha\beta}(\Pi_+ s_\alpha) = -ig\Pi_+\eta(\bar{s}^\beta\Gamma_5\Pi_+\eta), \tag{B.4}$$

$$R_\alpha^{\ \beta}(\bar{s}^\alpha\Pi_-) = ig\zeta^c\Pi_-(\bar{s}^\beta\Gamma_5\Pi_+\eta), \tag{B.5}$$

$$R_{\alpha\beta}(\bar{s}^\alpha\Pi_-) = -ig\zeta^c\Pi_-(\zeta^c\Pi_-\Gamma_5 s_\beta), \tag{B.6}$$

$$R^\alpha_{\ \beta}(\Pi_+ s_\alpha) = ig\Pi_+\eta(\zeta^c\Pi_-\Gamma_5 s_\beta) \tag{B.7}$$

$$\Rightarrow \zeta^c\Pi_-\Gamma_5 s_\beta = \bar{s}^\beta\Gamma_5\Pi_+\eta = 0, \tag{B.8}$$

where we have used $\bar{s}^\alpha\Pi_-\Gamma_5 s_\beta = 0$. Now we have $\mathcal{Q}_s B = \mathcal{Q}_s F = \mathcal{G}_s = 0$. The solution is

$$B = (R_\alpha q^\alpha + R^\alpha q_\alpha)(S_\beta q^\beta + S^\beta q_\beta), \quad R_\alpha s^\alpha = S_\alpha s^\alpha = \bar{s}^\alpha R_\alpha = \bar{s}^\alpha S_\alpha = 0, \tag{B.9}$$

$$\zeta^c\Pi_- = f(\tau)\bar{s}^1\Pi_-, \quad \Pi_+\eta = g(\tau)\Pi_+ s_1, \quad \bar{s}^\alpha\Gamma_{015}s_\beta = 0, \tag{B.10}$$

where $f(\tau)$ and $g(\tau)$ and are arbitrary functions.

## B.2   Supersymmetry enhancement

Using the explicit expression (49) for $L$, equation (56) can be decomposed into three equations:

$$\frac{2}{\bar{s}^\alpha\Pi_-\Gamma_5 s_\alpha}\mathcal{Q}_u\mathcal{Q}_s G_s = \partial_\tau G_u - i[L_{1/2}, G_u], \tag{B.11}$$

$$[G_s^2, G_u] = 0, \tag{B.12}$$

$$\{\mathcal{Q}_u G_s, G_s\} = \{\mathcal{Q}_s G_s, G_u\}. \tag{B.13}$$

It follows from (B.11) that $\zeta^c\Gamma_{015}u_\alpha = \bar{u}^\alpha\Gamma_{015}\eta = 0$ and from (B.12) that

$$G_u = KG_s = \begin{pmatrix} k^{(1)}I_N & 0 \\ 0 & k^{(2)}I_N \end{pmatrix}G_s. \tag{B.14}$$

or equivalently

$$i\zeta^c\Pi_-\Gamma_5 u_\alpha q^\alpha - iq_\alpha\bar{u}^\alpha\Gamma_5\Pi_+\eta = K(i\zeta^c\Pi_-\Gamma_5 s_\alpha q^\alpha - iq_\alpha\bar{s}^\alpha\Gamma_5\Pi_+\eta). \tag{B.15}$$

Then (B.13) leads to

$$\begin{aligned} &\{\mathcal{Q}_u G_s, G_s\} = \{\mathcal{Q}_s G_s, G_u\} \\ \Leftrightarrow &\mathcal{Q}_u G_s G_s = \mathcal{Q}_s G_s G_u \quad \text{and} \quad G_s\mathcal{Q}_u G_s = G_u\mathcal{Q}_s G_s \\ \Leftrightarrow &\mathcal{Q}_u G_u G_s = K\mathcal{Q}_s G_s KG_s \quad \text{and} \quad G_s\mathcal{Q}_u G_u = KG_s K\mathcal{Q}_s G_s \\ \Leftrightarrow &(\bar{u}^\alpha\Gamma_5\Pi_+ u_\alpha)\mathcal{Q}_s G_s = (\bar{s}^\beta\Gamma_5\Pi_+ s_\alpha)k^{(1)}k^{(2)}\mathcal{Q}_s G_s \\ \Leftrightarrow &k^{(1)}k^{(2)} = \frac{\bar{u}^\alpha\Gamma_5\Pi_+ u_\alpha}{\bar{s}^\alpha\Gamma_5\Pi_+ s_\alpha}. \end{aligned} \tag{B.16}$$

Now we are left with equations:

$$\zeta^{(i)c}\Pi_-\Gamma_5 u_\alpha = k^{(i)}\zeta^{(i)c}\Pi_-\Gamma_5 s_\alpha, \tag{B.17}$$

$$\bar{u}^\alpha\Gamma_5\Pi_+\eta^{(i)} = k^{(i)}\bar{s}^\alpha\Gamma_5\Pi_+\eta^{(i)}, \tag{B.18}$$

$$\zeta^{(i)c}\Gamma_{015}u_\alpha = \bar{u}^\alpha\Gamma_{015}\eta^i = 0. \tag{B.19}$$

We divide the discussion into two cases: the spinors $\zeta^{(i)}$ and $\eta^{(i)}$ satisfy $\zeta^{(i)c}\Gamma_{015}\zeta^{(i)} = \eta^{(i)c}\Gamma_{015}\eta^{(i)} = 0$ or not.

**Case 1:** $\zeta^{(i)c}\Gamma_{015}\zeta^{(i)} = \eta^{(i)c}\Gamma_{015}\eta^{(i)} = 0.$

Let us consider the linear equations involving $\zeta^{(1)c}$

$$\zeta^{(1)c}\Pi_-\Gamma_5 u_\alpha = k^{(1)}\zeta^{(1)c}\Pi_-\Gamma_5 s_\alpha, \tag{B.20}$$

$$\zeta^{(1)c}\Gamma_{015}u_\alpha = 0. \tag{B.21}$$

When $\zeta^{(1)c}\Gamma_{015}\zeta^{(1)} = 0$, the vector space spanned by $\{\zeta^{(1)}, \Gamma_{01}\zeta^{(1)}, \Gamma_{15}\zeta^{(1)}, \Gamma_{50}\zeta^{(1)}\}$ is two-dimensional. Let $V_{\zeta^{(1)}}$ denote the vector space spanned by $\{\zeta^{(1)}, \Gamma_{01}\zeta^{(1)}, \Gamma_{15}\zeta^{(1)}, \Gamma_{50}\zeta^{(1)}\}$. It is a subspace of the three-dimensional solution space of the equation $\zeta^{(1)c}\Gamma_{015}s_\alpha = 0$. Therefore $s_\alpha$ can be decomposed as

$$s_\alpha = c_\alpha s_{\perp\zeta^{(1)}} + s_{\alpha\|\zeta^{(1)}}, \tag{B.22}$$

where $s_{\alpha\|\zeta^{(1)}}$ is a vector in $V_{\zeta^{(1)}}$ and $s_{\perp\zeta^{(1)}}$ is a vector in the complementary transverse direction. One can use the inner product $s^\dagger s$ to define a unique $s_{\perp\zeta^{(1)}}$. The solution to (B.20) can be written as

$$u_\alpha = k^{(1)}c_\alpha s_{\perp\zeta^{(1)}} + u_{\alpha\|\zeta^{(1)}}. \tag{B.23}$$

So $k^{(1)}$ is a constant. Repeating similar analysis for the bottom-left block of the connection, $k^{(2)}$ is also constant and thus

$$k^{(1)}k^{(2)} = \frac{\bar{u}^\alpha\Gamma_5\Pi_+ u_\alpha}{\bar{s}^\alpha\Gamma_5\Pi_+ s_\alpha} = k^{(1)}\frac{\bar{u}^\alpha_{\|\zeta^{(1)}}\Gamma_5\Pi_+ c_\alpha s_{\perp\zeta^{(1)}}}{\bar{s}^\alpha_{\|\zeta^{(1)}}\Gamma_5\Pi_+ c_\alpha s_{\perp\zeta^{(1)}}} \tag{B.24}$$

is a constant. This leads to

$$\bar{u}^\alpha_{\|\zeta^{(1)}}\Gamma_i c_\alpha s_{\perp\zeta^{(1)}} = k^{(2)}\bar{s}^\alpha_{\|\zeta^{(1)}}\Gamma_i c_\alpha s_{\perp\zeta^{(1)}}, \tag{B.25}$$

for $i = 0, 1, 5$. The solution is

$$u_{\alpha\|\zeta^{(1)}} = k^{(2)}s_{\alpha\|\zeta^{(1)}} + c_\alpha u'_{\|\zeta^{(1)}}, \tag{B.26}$$

where $u'_{\|\zeta^{(1)}}$ is an arbitrary vector in $V_{\zeta^{(1)}}$. Finally, we get

$$u_\alpha = k^{(1)}c_\alpha s_{\perp\zeta^{(1)}} + k^{(2)}s_{\alpha\|\zeta^{(1)}} + c_\alpha u'_{\|\zeta^{(1)}}. \tag{B.27}$$

Now we consider linear equations involving $\zeta^{(2)c}$. There are two subcases:

(1) When $\zeta^{(1)}$ and $\zeta^{(2)}$ are proportional to the same vector, we find $k^{(1)} = k^{(2)}$. Furthermore, if $\zeta^{(i)}$ and $\eta^{(i)}$ are all proportional to a nonzero vector, the solution is

$$u_\alpha = k^{(1)}s_\alpha + c_\alpha u'_{\|\zeta^{(1)}} \tag{B.28}$$

and the Wilson loop is 3/16 BPS when $c_\alpha \neq 0$.[10] When $c_\alpha = 0$, we get $\bar{s}^\alpha\Pi_-\Gamma_5 s_\alpha = 0$ which is not consistent with our construction, as can be seen from (49).

(2) When $\zeta^{(1)}$ and $\zeta^{(2)}$ are linearly independent, we have either $V_{\zeta^{(1)}} = V_{\zeta^{(2)}}$ or $V_{\zeta^{(1)}} \cap V_{\zeta^{(2)}} = \{0\}$.

If $V_{\zeta^{(1)}} = V_{\zeta^{(2)}}$, $\zeta^{(1)c}\Gamma_{015}s_\alpha = \zeta^{(2)c}\Gamma_{015}s_\alpha = 0$ implies $s_\alpha \in V_{\zeta^{(1)}}$. In this case we get $\bar{s}^\alpha\Pi_-\Gamma_5 s_\alpha = 0$ which is not consistent with our construction.

---

[10]Taking $u'_{\|\zeta^{(1)},1}$ and $u'_{\|\zeta^{(1)},2}$ to be one basis of $V_{\zeta^{(1)}}$. $u_\alpha$ is in three-dimensional complex space spanned by $s_\alpha$, $c_\alpha u'_{\|\zeta^{(1)},1}$ $c_\alpha u'_{\|\zeta^{(1)},2}$ for each $\alpha$.

If $V_{\zeta^{(1)}} \cap V_{\zeta^{(2)}} = \{0\}$, the solution to $\zeta^{(1)c}\Gamma_{015}s_\alpha = \zeta^{(2)c}\Gamma_{015}s_\alpha = 0$ can be decomposed as

$$s_\alpha = c_\alpha^{(1)}s_{\|\zeta^{(1)}} + c_\alpha^{(2)}s_{\|\zeta^{(2)}}, \tag{B.29}$$

where $s_{\|\zeta^{(i)}} \in V_{\zeta^{(i)}}$. Therefore we get

$$u_\alpha = k^{(2)}c_\alpha^{(1)}s_{\|\zeta^{(1)}} + k^{(1)}c_\alpha^{(2)}s_{\|\zeta^{(2)}}. \tag{B.30}$$

The analysis for the $\eta$-equations is similar and we get

$$s_\alpha = c_\alpha s_{\perp\zeta^{(1)}} + s_{\alpha\|\zeta^{(1)}} = c_\alpha s_{\perp\eta^{(1)}} + s_{\alpha\|\eta^{(1)}}, \tag{B.31}$$

$$u_\alpha = k^{(1)}c_\alpha s_{\perp\zeta^{(1)}} + k^{(2)}s_{\alpha\|\zeta^{(1)}} + u'_{\|\zeta^{(1)}} = k^{(1)}c_\alpha s_{\perp\eta^{(1)}} + k^{(2)}s_{\alpha\|\eta^{(1)}} + u'_{\|\eta^{(1)}}. \tag{B.32}$$

Let us consider the case when $\zeta^{(1)}$ is not proportional to $\eta^{(1)}$. Now if $V_{\zeta^{(1)}} = V_{\eta^{(1)}}$, $\zeta^{(1)c}\Gamma_{015}s_\alpha = \eta^{(1)c}\Gamma_{015}s_\alpha = 0$ leads to $\bar{s}^\alpha\Pi_-\Gamma_5 s_\alpha = 0$. On the other hand, if $V_{\zeta^{(1)}} \cap V_{\eta^{(1)}} = \{0\}$, equations (B.32) and (B.31) lead to $u_\alpha = k^{(1)}s_\alpha$. Therefore supersymmetry enhancement is possible only when $\zeta^{(1)} \propto \eta^{(1)}$, $\zeta^{(2)} \propto \eta^{(2)}$ and $V_{\zeta^{(1)}} \cap V_{\zeta^{(2)}} = \{0\}$. In this case the Wilson loop is 1/8-BPS and the preserved supercharges can be parameterized by

$$u_\alpha = k^{(2)}c_\alpha^{(1)}s_{\|\zeta^{(1)}} + k^{(1)}c_\alpha^{(2)}s_{\|\zeta^{(2)}}. \tag{B.33}$$

Now $u_\alpha$ is in the two-dimensional complex space spanned by $c_\alpha^{(1)}s_{\|\zeta^{(1)}}$ and $c_\alpha^{(2)}s_{\|\zeta^{(2)}}$ for each $\alpha$.

**Case 2:** At least one of the spinors $\zeta^{(i)}$ and $\eta^{(i)}$ satisfies $\chi^c\Gamma_{015}\chi \neq 0$, $\chi = \zeta^{(i)}$ or $\eta^{(i)}$.

Without loss of generality, we assume $\zeta^{(1)c}\Gamma_{015}\zeta^{(1)} \neq 0$ and thus $\zeta^{(1)c}\Gamma_0$, $\zeta^{(1)c}\Gamma_1$, $\zeta^{(1)c}\Gamma_5$ and $\zeta^{(1)c}\Gamma_{015}$ are linearly independent. We can decompose a spinor $u$ satisfying $\zeta^{(1)c}\Gamma_{015}u = 0$ as

$$u = U_1\Gamma_{01}\zeta^{(1)} + U_2\Gamma_{15}\zeta^{(1)} + U_3\Gamma_{50}\zeta^{(1)}. \tag{B.34}$$

Therefore we can associate a spinor $u$ with a three-dimensional vector $V(u) = (U_1, U_2, U_3)$. We find

$$V(s) \cdot V(u) \equiv \sum_{i=1}^{3} V_i(s)V_i(u) = \frac{s^c\Gamma_{015}u}{\zeta^{(1)c}\Gamma_{015}\zeta^{(1)}}, \tag{B.35}$$

$$\frac{\zeta^{(1)c}\Pi_-\Gamma_5 u}{\zeta^{(1)c}\Gamma_{015}\zeta^{(1)}} = V(u) \cdot Z, \tag{B.36}$$

where $Z = \frac{1}{2}(1, -i\sin\tau, i\cos\tau)$. Denoting $S_\alpha = (S_{\alpha 1}, S_{\alpha 2}, S_{\alpha 3}) = V(s_\alpha)$ and $U_\alpha = V(u_\alpha)$, equation (B.20) is equivalent to

$$S_1 \cdot Z\, U_2 \cdot Z - S_2 \cdot Z\, U_1 \cdot Z = 0. \tag{B.37}$$

From the coefficients of $1, \sin\tau, \cos\tau, \sin 2\tau, \cos 2\tau$, we get

$$2U_{11}S_{21} - 2S_{11}U_{21} - U_{12}S_{22} + S_{12}U_{22} - U_{13}S_{23} + S_{13}U_{23} = 0, \tag{B.38}$$

$$-U_{11}S_{22} + S_{11}U_{22} - U_{12}S_{21} + S_{12}U_{21} = 0, \tag{B.39}$$

$$U_{11}S_{23} - S_{11}U_{23} + U_{13}S_{21} - S_{13}U_{21} = 0, \tag{B.40}$$

$$U_{12}S_{23} - S_{12}U_{23} + U_{13}S_{22} - S_{13}U_{22} = 0, \tag{B.41}$$

$$U_{12}S_{22} - S_{12}U_{22} - U_{13}S_{23} + S_{13}U_{23} = 0. \tag{B.42}$$

Viewing them as linear equations of $U_\alpha$ and computing the rank of the matrix of the coefficients, we find when $\det S_\alpha \cdot S_\beta \neq 0$, the solution is $U_\alpha = k^{(1)}S_\alpha$ and thus $u_\alpha = k^{(1)}s_\alpha$ and $k^{(1)}$ is a constant.

When $\det S_\alpha \cdot S_\beta = 0$, there is a vector $E_+ = a_1 S_1 + a_2 S_2$ such that $E_+ \cdot S_\alpha = 0$. We can further choose a basis of vectors $\{E_+, E_-, E_0\}$ such that

$$E_+ \cdot E_- = 2 E_0 \cdot E_0 = 2, E_0 \cdot E_\pm = E_\pm \cdot E_\pm = 0. \tag{B.43}$$

Then $S_\alpha$ can be decomposed as

$$S_\alpha = M_\alpha{}^+ E_+ + M_\alpha{}^0 E_0. \tag{B.44}$$

The solution of (B.20) can be written as

$$U_\alpha = M_\alpha{}^+ (\kappa_1 E_+ + \kappa_2 E_0) + M_\alpha{}^0 (\kappa_1 E_0 - \kappa_2 E_-). \tag{B.45}$$

For each $\alpha$, $U_\alpha$ is in the two dimensional complex space spanned by $S_\alpha$ and $M_\alpha{}^+ E_0 - M_\alpha{}^0 E_-$. We get

$$k^{(1)} = \frac{Z \cdot (\kappa_1 E_+ + \kappa_2 E_0)}{Z \cdot E_+}. \tag{B.46}$$

Using (B.16), we find

$$k^{(1)} k^{(2)} = \frac{\bar{u}^\alpha \Gamma_5 \Pi_+ u_\alpha}{\bar{s}^\alpha \Gamma_5 \Pi_+ s_\alpha} = \frac{\epsilon^{ijk} Z_i U_{1j} U_{2k}}{\epsilon^{ijk} Z_i S_{1j} S_{2k}} = k^{(1)2}, \tag{B.47}$$

where one can decompose $Z$ over the basis $\{E_+, E_-, E_0\}$ and use $Z \cdot Z = 0$ to derive the last equality. Therefore we get $k^{(1)} = k^{(2)}$.

When $\zeta^{(2)c} \Gamma_{015} \zeta^{(2)} \neq 0$ or $\eta^{(1,2)c} \Gamma_{015} \eta^{(1,2)} \neq 0$, we get similar results. So in the case, when $\zeta^{(i)}$ and $\eta^{(i)}$ are all proportional to a nonzero bosonic spinor $\chi$ satisfying $\chi^c \Gamma_{015} \chi \neq 0$ and

$$(s_1^c \Gamma_{015} s_1)(s_2^c \Gamma_{015} s_2) - (s_1^c \Gamma_{015} s_2)^2 = 0, \tag{B.48}$$

the Wilson loop is 1/8-BPS. Otherwise, one can show the Wilson loop is 1/16-BPS, only the supercharges proportional to $\mathcal{Q}_s$ are preserved.

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
