# Peer review of "Fermionic Bogomolnyi-Prasad-Sommerfield Wilson loops in four-dimensional $\mathcal{N}=2$ superconformal gauge theories"

_SciPost Physics, doi:SciPost Phys. 14, 008 (2023)_

## Round 2 · Referee Report · Anonymous (Referee 1) · 2022-7-24

Report

The results are relevant but the way they are presented can be improved. I suggest a revision before publishing. I attach a PDF with detailed comments and suggestions that I believe would increase the quality of the paper.

Attachment

  • validity: -
  • significance: -
  • originality: -
  • clarity: -
  • formatting: -
  • grammar: -

Author:  Jun-Bao Wu  on 2022-09-20  [id 2832]

(in reply to Report 2 on 2022-07-24)

We thank the referee for his/her report and careful reading of our manuscript.

1) We thank the referee for pointing out many typos, which we have corrected in the revised version.

2) We added a paragraph on page 3 and a footnote on page 4 to address the referee's comments on our introduction. We also added a paragraph on page 4 to stress that this is the first construction of Drukker-Trancanelli type fermionic BPS Wilson loops in 4d.

3) We added a sentence above (48) to mention how the $G_s^2$ piece is obtained in section 2.3.

4) The number of preserved supercharges is equal to the number of free parameters in $u_\alpha$. In another word, it is just the dimension of the linear space of the preserved supercharges. Detailed clarifications were added on page 12.

5) It should be $QF$ above (44), as pointed out by the referee.

6) We added a paragraph at the end of section 4 to discuss the consequences of scale invariance breaking due to fermions in the WL.

7) We changed the order of B.1 and B.2.

---

## Round 2 · Referee Report · Anonymous (Referee 2) · 2022-8-8

Strengths

1) the authors apply to 4-dimensional theories an algorithm to construct fermionic BPS Wilson loops that was previously only used in the case of 3-dimensional theories, so this novelty is a strength of the paper

2) the methodology used in the construction is sound

3) the computations, to the extent I could check them, are correct

4) the paper is clearly written, albeit quite technical

Weaknesses

1) some novelties appearing in the 4-dimensional case should be further discussed: for example, the appearance of dimensionful couplings in the definition of the fermionic Wilson loops

2) the discussion of what is expected to happen at the quantum level could be improved

3) maybe some more details about the computation of the cohomological equivalence could be given (for example, the expression for V at the lowest orders in the expansion)

4) the paper is quite technical, albeit clearly written

Report

The authors apply to 4-dimensional theories an algorithm to construct novel supersymmetric Wilson loops, that was previously discovered in the context of 3-dimensional quiver theories, like ABJ(M) and N=4 Chern-Simons-matter theories. This is an interesting attempt, resulting in a few novel Wilson loops defined in terms of superconnections. An obvious difference with the 3-dimensional case is that the couplings to the fields appearing in this superconnection are dimensionful, as can be immediately seen by dimensional analysis. The construction is quite interesting and the computations are correct, for the extent I could check them.

Requested changes

1) I would improve the discussion of the first 3 points mentioned above in the "Weaknesses" section.

  • validity: high
  • significance: good
  • originality: good
  • clarity: good
  • formatting: excellent
  • grammar: good

Author:  Jun-Bao Wu  on 2022-09-20  [id 2831]

(in reply to Report 1 on 2022-08-08)

We thank the referee very much for his/her valuable comments. We address the requests:

1) We added a paragraph at the end of section 4 to discuss the consequences of the dimensionful couplings.

2) We added a comment about the framing issue at the end of the second paragraph in section 4 which is related to the perturbative computation of the WL vev.

3) We gave the first few order result of V in Eq (53). The complete construction of V can be performed following the procedure in appendix D of 1506.06192.

---

## Round 3 · Referee Report · Anonymous (Referee 1) · 2022-10-2

Report

The authors have addressed my comments and I think the
paper is now much improved. I think it can be published.

---

## Round 3 · Author Response

Dear Editors,

Let us thank the referees for their careful reading of the manuscript and valuable comments. We resubmit a modified version of our manuscript after making the changes suggested. We have attached the replies to issues raised by the referees and corresponding changes below.

Sincerely yours,

Hao Ouyang and Jun-Bao Wu

---

## Round 3 · List of Changes

Reply to Anonymous Report 1:

We thank the referee very much for his/her valuable comments. We address the requests:

1) We added a paragraph at the end of section 4 to discuss the consequences of the dimensionful couplings.

2) We added a comment about the framing issue at the end of the second paragraph in section 4 which is related to the perturbative computation of the WL vev.

3) We gave the first few order result of V in Eq (53). The complete construction of $V$ can be performed following the procedure in appendix D of 1506.06192.

Reply to Anonymous Report 2:

We thank the referee for his/her report and careful reading of our manuscript.

1) We thank the referee for pointing out many typos, which we have corrected in the revised version.

2) We added a paragraph on page 3 and a footnote on page 4 to address the referee's comments on our introduction. We also added a paragraph on page 4 to stress that this is the first construction of Drukker-Trancanelli type fermionic BPS
Wilson loops in 4d.

3) We added a sentence above (48) to mention how the G_s^2
piece is obtained in section 2.3.

4) The number of preserved supercharges is equal to the number of free parameters in u_\alpha. In another word, it is just the dimension of the linear space of the preserved supercharges. Detailed clarifications were added on page 12.

5) It should be QF above (44), as pointed out by the referee.

6) We added a paragraph at the end of section 4 to discuss the consequences of scale invariance breaking due to fermions in the WL.

7) We changed the order of B.1 and B.2.

---

## Editorial Decision

published